# FlashBlock: Attention Caching for Efficient Long-Context Block Diffusion

Zhuokun Chen [1] [*]   Jianfei Cai [1]   Bohan Zhuang [2]

## Abstract

Generating long-form content, such as minute-long videos and extended texts, is increasingly important for modern generative models. Block diffusion improves inference efficiency via KV caching and block-wise causal inference and has been widely adopted in diffusion language models and video generation. However, in long-context settings, block diffusion still incurs substantial overhead from repeatedly computing attention over a growing KV cache. We identify an underexplored property of block diffusion: cross-step redundancy of attention within a block. Our analysis shows that attention outputs from tokens outside the current block remain largely stable across diffusion steps, while block-internal attention varies significantly. Based on this observation, we propose `FlashBlock`, a cached block-external attention mechanism that reuses stable attention output, reducing attention computation and KV cache access without modifying the diffusion process. Moreover, `FlashBlock` is orthogonal to sparse attention and can be combined as a complementary residual reuse strategy, substantially improving model accuracy under aggressive sparsification. Experiments on diffusion language models and video generation demonstrate up to $1.44\times$ higher token throughput and up to $1.6\times$ reduction in attention time, with negligible impact on generation quality. Project page: https://caesarhhh.github.io/FlashBlock/.

## 1. Introduction

Diffusion models have shown strong performance in both language modeling and video generation by iteratively refining noisy representations over multiple steps. However, their iterative nature leads to high inference cost, especially in long-context settings. To address this issue, recent work has proposed *block diffusion*, which combines diffusion with autoregressive generation by introducing KV caching and performing block-by-block causal inference (Arriola et al., 2025a; Hoogeboom et al., 2022). By refining a block of tokens at each diffusion step while conditioning on cached representations from previous blocks, block diffusion enables efficient and variable-length generation.

Despite improved per-step efficiency, block diffusion remains inefficient in *long-context* settings. At each diffusion step, attention is computed over all previously generated tokens, requiring repeated access to an ever-growing KV cache. As context length increases, the combined cost of attention computation and KV cache access rapidly dominates inference latency. This effect is particularly pronounced in diffusion language models with long sequences and in video diffusion models with extended temporal horizons, where attention is repeatedly evaluated across many diffusion steps. Consequently, reducing both attention computation and KV cache access is critical for further improving the efficiency of block diffusion inference.

A key but underexplored property of block diffusion is the presence of *cross-step redundancy* within a block: during denoising, attention is recomputed at successive diffusion steps over highly similar token representations. To examine this behavior, we empirically analyze attention outputs across diffusion steps, as shown in Figure 1. We observe a clear separation between block-internal and block-external attention. While block-internal attention varies substantially as tokens within the block are actively updated, attention contributions from tokens outside the current block remain largely stable across adjacent diffusion steps. This finding indicates that repeatedly recomputing block-external attention is largely redundant, and that reusing these stable attention results can significantly reduce attention computation and KV cache access during block diffusion inference.

Based on this observation, we propose a **block-external attention reusing** mechanism for block diffusion. Our method caches attention outputs corresponding to tokens outside the current block and reuses them in subsequent diffusion steps, while recomputing only the attention within the current block. The full attention output is obtained by combining cached block-external attention with newly computed block-

---

[1]Monash University [2]Zhejiang University. Correspondence to: Zhuokun Chen <caesard216@gmail.com>, Bohan Zhuang <bohan.zhuang@gmail.com>.

*Proceedings of the $43^{rd}$ International Conference on Machine Learning*, Seoul, South Korea. PMLR 306, 2026. Copyright 2026 by the author(s).

internal attention, without modifying the underlying diffusion process. This design substantially reduces KV cache access and attention computation overhead, making block diffusion more efficient in long-context scenarios. Moreover, our approach is orthogonal to sparse attention and can be seamlessly combined as a complementary caching strategy: it reuses the residual attention contribution from unselected tokens across diffusion steps, effectively compensating for information loss introduced by sparsification and improving generation quality.

We evaluate our method primarily on diffusion language models, with additional experiments on video generation diffusion models. Experimental results show that our approach achieves up to $1.44\times$ inference speedup on diffusion language models and up to $1.6\times$ reduction in attention time, while incurring only negligible impact on generation quality across tasks. We further show that our method can be combined with sparse attention to mitigate the quality degradation introduced by aggressive sparsification at the same attention density.

Overall, our contributions are summarized as follows:

- We present an empirical study of attention behavior in block diffusion, revealing strong cross-step stability in block-external attention across diffusion steps, in contrast to the highly variable block-internal attention.

- We propose FlashBlock, a block-external attention caching mechanism that exploits this cross-step redundancy to reduce attention computation and KV cache access during inference.

- We show that FlashBlock is compatible with sparse attention methods and can be combined as a complementary residual reuse strategy to alleviate the quality degradation caused by aggressive sparsification.

- We demonstrate that FlashBlock substantially accelerates inference on diffusion language models and video diffusion models, with only negligible impact on generation quality.

## 2. Related Work

**Block diffusion.** Diffusion-based sequence models have been explored as an alternative to autoregressive decoding due to their potential for parallel token generation (Song et al.; Ho et al., 2020). Early diffusion language models include both continuous-space formulations (Li et al., 2022) and discrete masked diffusion for conditional generation (Gong et al.; Austin et al., 2021), while recent large-scale diffusion LLMs demonstrate competitive performance at scale (e.g., LLaDA (Nie et al., 2025), Dream (Ye et al.,

2025)). However, standard diffusion inference recomputes attention over the full sequence at every denoising step, leading to high latency and poor support for variable-length generation and KV caching. Block diffusion mitigates these issues by introducing block-by-block causal inference with KV caching, reusing historical context across blocks to improve per-step efficiency and enable flexible-length decoding (Arriola et al., 2025a). This paradigm has been adopted in efficient diffusion LLM systems (e.g., Fast-dLLM (Wu et al., 2025; Arriola et al., 2025b)) and extended to long-horizon video generation, where models condition each chunk on prior chunks via block-causal mechanisms (e.g., CausVid (Yin et al., 2025), BlockVid (Zhang et al., 2025b)). In parallel, several works accelerate diffusion LLM inference by reusing or approximating KV states across denoising steps (Hu et al., 2025; Ma et al., 2025), highlighting KV-related computation as a key efficiency bottleneck. Nevertheless, these approaches are designed for standard diffusion settings and do not exploit the structural properties of block diffusion. In long-context regimes, block diffusion still recomputes attention over an ever-growing KV cache at each denoising step within a block, leaving substantial cross-step redundancy in block-causal attention unexploited.

**Sparse attention.** Sparse attention is a classical direction for scaling transformers to long sequences by reducing the number of attended keys per query (Chen et al., 2023; Beltagy et al., 2020). Recently, long-context LLM inference has emphasized *KV-cache-aware* sparsification and selection to reduce memory movement during decoding, including query-aware page selection (Tang et al., 2024) and token-level retention/eviction based on attention concentration (Zhang et al., 2023; Adnan et al., 2024). In the context of diffusion language models, SparseD (Wang et al., 2025b) further adapts sparse attention to diffusion inference by identifying head-specific sparse patterns and reusing them across denoising steps, while applying full attention in early steps to preserve generation quality (Wang et al., 2025b). Existing sparse attention methods exploit sparsity within individual attention calls but do not capture the cross-step redundancy inherent to block diffusion, where attention is repeatedly recomputed across diffusion steps on similar representations.

## 3. Empirical Insights

We begin by empirically examining how attention behaves across diffusion steps in diffusion language models. Specifically, we analyze the cross-step stability of attention contributions from tokens inside and outside the current block by comparing attention outputs across adjacent diffusion steps. A corresponding visualization and analysis for video diffusion models is provided in the appendix.

**Stable block-external attention.** We first analyze atten-

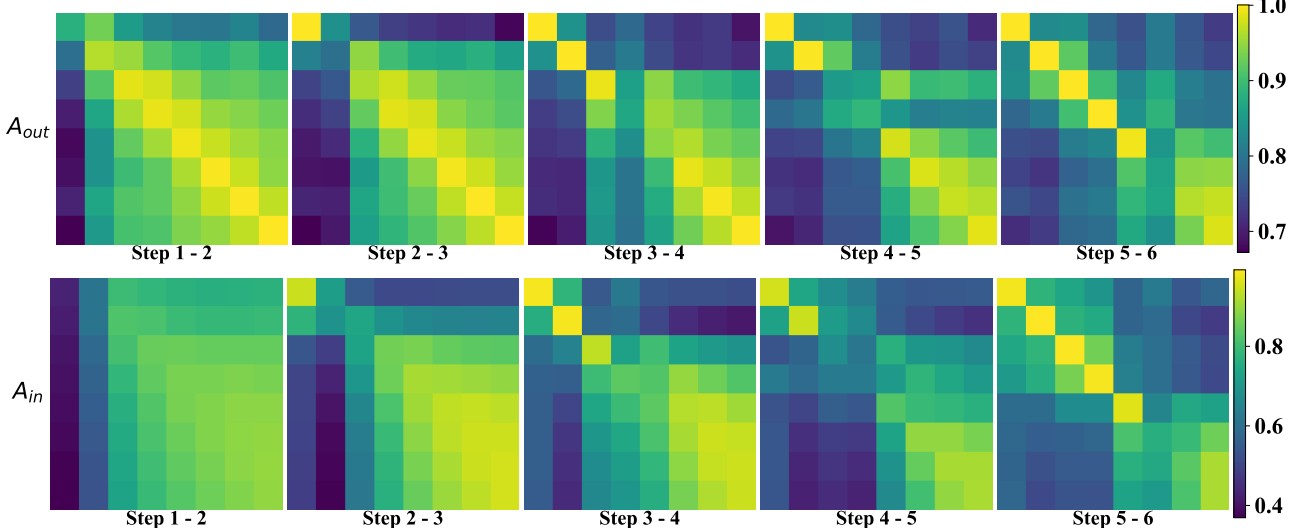

*Figure 1.* **Cross-step stability of block-external vs. block-internal attention across diffusion steps.** Visualization of attention similarity across diffusion steps for the same block at *layer 3* of Trado-8B-Thinking. We compute the similarity of attention outputs between each diffusion step and its subsequent step within a denoising block. In each heatmap, the x-axis corresponds to token indices within the current block at step $s$, and the y-axis corresponds to token indices within the same block at step $s+1$; diagonal entries therefore represent the similarity of the same token across adjacent diffusion steps. The top row shows block-external attention ($A_{\text{out}}$), and the bottom row shows block-internal attention ($A_{\text{in}}$), across multiple diffusion steps. Brighter colors indicate higher similarity. Across steps, $A_{\text{out}}$ consistently exhibits higher similarity and more coherent structure, indicating strong cross-step stability, whereas $A_{\text{in}}$ varies substantially across steps.

tion contributions from tokens outside the current block. For a fixed attention head, we compute the similarity of block-external attention outputs $A_{\text{out}}$ between consecutive diffusion steps. Figure 1 (top row) visualizes these similarities across multiple diffusion steps. Each heatmap corresponds to one diffusion step transition, where brighter colors indicate higher similarity. Across all steps, block-external attention exhibits consistently high similarity and clear structural alignment, indicating that attention contributions from previously generated tokens remain largely unchanged as diffusion progresses. This strong cross-step stability suggests that recomputing block-external attention at every diffusion step is largely redundant.

**Variable block-internal attention.** In contrast, block-internal attention outputs $A_{\text{in}}$ show substantially more variation across diffusion steps. As shown in the bottom row of Figure 1, the similarity patterns of block-internal attention differ noticeably from step to step and lack the consistent structure observed in $A_{\text{out}}$. This variability arises because tokens inside the current block are actively updated during block diffusion, either through refinement or unmasking, leading to rapidly changing interactions among block-internal tokens.

# 4. Methodology

Motivated by the above observations, we introduce a method that exploits the cross-step stability of block-external attention to reduce redundant computation and KV cache access.

In this section, we first formalize the attention decomposition in block diffusion, then present an efficient inference procedure based on block-external attention caching, and finally describe a training strategy to further align sparse and dense attention behaviors.

## 4.1. Preliminaries

We consider block diffusion models that perform block-by-block causal inference over a sequence of tokens. Let $x \in \mathbb{R}^{N \times d}$ denote the token representations at a given diffusion step, where $N$ is the sequence length and $d$ is the hidden dimension. At each diffusion step $s$, the model updates the representations of a contiguous block of tokens while keeping the remaining tokens unchanged. Attention is computed using cached key–value (KV) representations from previously generated tokens.

For a query token $i$, the scaled dot-product attention score with key token $j$ is defined as

$$s_{ij} = \frac{q_i^\top k_j}{\sqrt{d}}, \tag{1}$$

where $q_i \in \mathbb{R}^d$ is the query vector, and $k_j, v_j \in \mathbb{R}^d$ are the key and value vectors, respectively. The attention output is given by

$$Z_i = \sum_j e^{s_{ij}}, \quad U_i = \sum_j e^{s_{ij}} v_j, \quad a_i = \frac{U_i}{Z_i}. \tag{2}$$

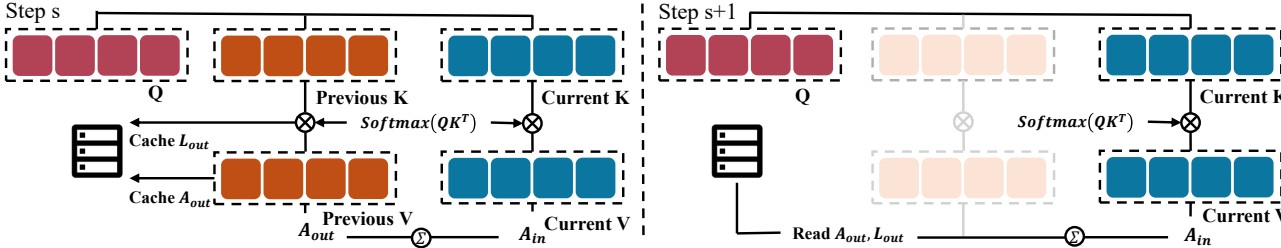

*Figure 2.* **Block-external attention caching for block diffusion.** At each diffusion step, block diffusion updates a contiguous block of tokens. Our method caches attention contributions from block-external tokens and reuses them across steps, recomputing attention only within the current block. Block-internal and block-external attention are combined via log-space aggregation, reducing computation and memory I/O in long-context settings.

### 4.2. Block-Causal Attention Caching

**Attention Decomposition.** Let $\mathcal{J}_{\text{in}}$ denote the set of key indices inside the current block, and $\mathcal{J}_{\text{out}}$ denote the set of key indices outside the current block. We define the block-internal and block-external attention statistics as

$$Z_{i,\text{in}} = \sum_{j \in \mathcal{J}_{\text{in}}} e^{s_{ij}}, \qquad U_{i,\text{in}} = \sum_{j \in \mathcal{J}_{\text{in}}} e^{s_{ij}} v_j, \qquad (3)$$

$$Z_{i,\text{out}} = \sum_{j \in \mathcal{J}_{\text{out}}} e^{s_{ij}}, \qquad U_{i,\text{out}} = \sum_{j \in \mathcal{J}_{\text{out}}} e^{s_{ij}} v_j. \qquad (4)$$

The full attention output can be written as

$$a_i = \frac{U_{i,\text{out}} + U_{i,\text{in}}}{Z_{i,\text{out}} + Z_{i,\text{in}}}. \qquad (5)$$

**Block-external attention caching.** Block diffusion repeatedly computes attention across adjacent diffusion steps on highly similar representations. For tokens outside the current block, both the key–value representations and their attention contributions evolve slowly across steps. We therefore cache the block-external attention output and its normalizer at diffusion step $s$:

$$A_{\text{out}}^s = \frac{U_{\text{out}}^s}{Z_{\text{out}}^s}, \quad L_{\text{out}}^s = \log Z_{\text{out}}^s. \qquad (6)$$

The cached tensors depend only on the number of query tokens in the current block and are independent of the context length.

**Block-internal attention recomputation.** At the subsequent diffusion step $s+1$, attention over block-internal tokens is always recomputed:

$$A_{\text{in}}^{s+1} = \frac{U_{\text{in}}^{s+1}}{Z_{\text{in}}^{s+1}}, \quad L_{\text{in}}^{s+1} = \log Z_{\text{in}}^{s+1}. \qquad (7)$$

**Log-space attention composition.** When cached block-external attention is reused, the full attention output is obtained by composing block-external and block-internal attention in log space. Following the FlashAttention (Dao

et al., 2022) implementation, we perform the composition by subtracting the maximum log-normalizer $m$ from both the numerator and denominator to improve numerical stability under half-precision arithmetic.

Let

$$m = \max\left(L_{\text{out}}^s, L_{\text{in}}^{s+1}\right). \qquad (8)$$

The attention output at step $s+1$ is computed as

$$A_{\text{full}}^{s+1} = \frac{e^{L_{\text{out}}^s - m} A_{\text{out}}^s + e^{L_{\text{in}}^{s+1} - m} A_{\text{in}}^{s+1}}{e^{L_{\text{out}}^s - m} + e^{L_{\text{in}}^{s+1} - m}}. \qquad (9)$$

For diffusion steps where block-external attention is recomputed, the same formulation applies with updated $(A_{\text{out}}^{s+1}, L_{\text{out}}^{s+1})$.

**Selective reuse of block-external attention.** In discrete block diffusion models, token representations within a block may be updated abruptly at a diffusion step. Let $M^{s+1}$ denote the number of updated tokens in the current block at step $s+1$. We introduce a threshold $\tau$ to control whether block-external attention should be reused or recomputed. If the current block is processed for the first time, block-external attention is computed using the standard attention formulation and cached. Otherwise, if cached block-external attention is available and $M^{s+1} < \tau$, we directly reuse the cached block-external attention $(A_{\text{out}}^s, L_{\text{out}}^s)$ without accessing the KV cache. When $M^{s+1} \geq \tau$, block-external attention is recomputed to ensure correctness and the cache is updated accordingly.

For video diffusion models, we further adopt a head-wise selective reuse strategy. Before inference, we estimate the similarity of block-external attention between adjacent diffusion steps for each attention head using a small set of samples. During inference, cached block-external attention is reused only for heads whose estimated similarity exceeds a threshold $\gamma$, while block-external attention is recomputed for the remaining heads. This design accounts for the higher cross-step variability in video generation and avoids incorrect reuse when attention patterns change across diffusion

steps.

**Compatibility with Sparse Attention.** Our formulation naturally extends to sparse attention mechanisms. Sparse attention computes attention over a selected subset of keys for each query, while ignoring or approximating the remaining context. When combined with sparse attention, the previously defined partition $(\mathcal{J}_{\text{in}}, \mathcal{J}_{\text{out}})$ admits a natural interpretation: $\mathcal{J}_{\text{in}}$ corresponds to the set of keys selected by the sparse attention policy, over which attention is explicitly computed, while $\mathcal{J}_{\text{out}}$ corresponds to the complementary set of unselected keys.

Under this interpretation, sparse attention operates on the block-internal component $\mathcal{J}_{\text{in}}$, whereas block-external attention captures the residual contribution from $\mathcal{J}_{\text{out}}$. The same attention decomposition and log-space composition described in Section 4.2 therefore apply without modification. Our block-external attention caching mechanism can be directly applied on top of sparse attention by caching and reusing the residual attention term associated with $\mathcal{J}_{\text{out}}$ across diffusion steps, while recomputing sparse attention over $\mathcal{J}_{\text{in}}$. This makes our method complementary to attention sparsification and enables seamless integration with existing sparse attention mechanisms.

### 4.3. Inference Algorithm

---

**Algorithm 1** Inference Procedure with Block-External Attention Caching at step $s + 1$

---

**Input:** Query, key, value representations at diffusion step $s$, current block index set $\mathcal{J}_{\text{in}}$, cached block-external attention $(A_{\text{out}}^{s}, L_{\text{out}}^{s})$
**Output:** Attention output $A_{\text{full}}^{s+1}$ at diffusion step $s + 1$

1:  // Compute block-internal attention
2: Compute attention scores $s_{ij}^{s+1}$ for $j \in \mathcal{J}_{\text{in}}$
3: Compute block-internal normalizer $Z_{\text{in}}^{s+1}$ and weighted sum $U_{\text{in}}^{s+1}$
4: Obtain block-internal attention output $A_{\text{in}}^{s+1}$ and log-normalizer $L_{\text{in}}^{s+1}$
5:  // Compose attention outputs in log space
6: **for** each query token $i$ **do**
7:   $m \leftarrow \max(L_{i,\text{out}}^{s}, L_{i,\text{in}}^{s+1})$
8:   Combine block-external and block-internal attention (Eq. (5))
9: **end for**
10:  // Update block-external attention cache
11: Set $A_{\text{out}}^{s+1} \leftarrow A_{\text{full}}^{s+1}$ for tokens in $\mathcal{J}_{\text{out}}$
12: Set $L_{\text{out}}^{s+1} \leftarrow \log Z_{\text{out}}^{s+1}$
13: **return** $A_{\text{full}}^{s+1}$

---

### 4.4. Reuse-Aware Distillation

While the proposed inference procedure can be applied without modifying model parameters, directly introducing block-external attention caching may lead to distributional mismatch in diffusion language models. This issue mainly arises in discrete diffusion language models, where tokens are progressively unmasked during denoising, causing abrupt changes in attention contributions across diffusion steps. Such behavior can violate the stability assumption underlying attention reuse and degrade generation quality.

To mitigate this effect, we introduce a reuse-aware distillation strategy for dLLMs. This adaptation is optional and lightweight: it is implemented with LoRA adapters while keeping the base model weights frozen, and therefore preserves the inference-time nature of the proposed attention-reuse mechanism. We adopt a teacher–student formulation, where the teacher is a frozen model using dense attention and the student employs block-external attention caching. Let $p_{\text{teacher}}$ denote the output distribution of the teacher, and $p_{\text{reuse}}$ denote the output distribution of the student under attention reuse. We minimize

$$\mathcal{L}_{\text{reuse}} = \text{KL}(p_{\text{teacher}} \| p_{\text{reuse}}). \quad (10)$$

To stabilize training, we additionally compute the student output using dense attention, denoted as $p_{\text{dense}}^{\text{student}}$, and minimize

$$\mathcal{L}_{\text{reg}} = \text{KL}\big(p_{\text{teacher}} \| p_{\text{dense}}^{\text{student}}\big). \quad (11)$$

The final objective is

$$\mathcal{L} = \mathcal{L}_{\text{reuse}} + \lambda \mathcal{L}_{\text{reg}}, \quad (12)$$

where $\lambda$ controls the strength of the regularization term.

**Computation and memory complexity.** Let $B$ denote the block size (i.e., the number of query tokens updated at each diffusion step) and $N$ denote the total context length. In standard block diffusion, attention computation at each step requires accessing the full KV cache, resulting in $O(BN)$ KV cache accumulation and memory access. Our method is implemented on top of FlashAttention, which computes attention by streaming over keys while maintaining running accumulators for the weighted sum and the log-normalizer. During this streaming process, when the key index reaches the block boundary, we copy the current accumulated block-external attention output $A_{\text{out}}$ and log-normalizer $L_{\text{out}}$. This operation incurs a constant-time copy per query token and does not require additional passes over the KV cache.

At subsequent diffusion steps, attention computation is restricted to block-internal keys, reducing KV access and accumulation to $O(B^2)$ per step. The log-space composition of block-external and block-internal attention involves only elementwise operations over block queries and therefore

*Table 1.* Main results on diffusion language model benchmarks with $\tau = 2$. We report accuracy (%) for mathematical reasoning tasks and pass@1 (%) for coding benchmarks. Throughput (TPS) is measured as tokens per second with batch size 128 under an 800k context length. All results are evaluated under identical decoding settings.

| Method | Block Size | TPS | Mathematical Reasoning | | | Coding | | | |
| --- | --- | --- | --- | --- | --- | --- | --- | --- | --- |
| | | | GSM8K | MATH500 | AIME | MBPP | HumanEval | LiveCodeBench-V2 | LiveBench |
| Trado-8B-Thinking | 4 | 312 | 93.25 | 86.00 | 33.33 | 25.60 | 50.61 | 36.79 | 32.03 |
| Trado-8B-Thinking + Ours | 4 | 451 | 93.12 | 85.80 | 33.33 | 33.60 | 51.22 | 35.64 | 31.25 |
| Trado-8B-Thinking | 8 | 532 | 91.74 | 82.00 | 26.67 | 29.00 | 54.27 | 27.01 | 30.47 |
| Trado-8B-Thinking + Ours | 8 | 674 | 90.22 | 81.80 | 26.67 | 32.00 | 53.66 | 26.42 | 29.34 |

has $O(B)$ computational complexity. In terms of memory overhead, the cached block-external attention outputs and log-normalizers scale linearly with the block size. *Compared to the KV cache, whose size grows with the full context length $N$, the additional storage required by our method is negligible.* Overall, our approach significantly reduces attention computation and KV cache access in long-context settings while introducing minimal computational and memory overhead.

## 5. Experiments

**Implementation details.** In reuse-aware distillation, we employ a lightweight LoRA-based training scheme, inserting LoRA (Hu et al., 2021) adapters into the query, key, and value projection layers with rank 32. The student model is trained for 5,000 iterations on the DAPO-Math-17K dataset (Yu et al., 2025). This optional adaptation takes approximately three days on two NVIDIA A100 GPUs in our implementation and does not modify the base model weights. During training, we randomly roll out 1,000–4,000 tokens and perform distillation on the subsequent block by matching the student outputs to a frozen dense-attention teacher. The regularization coefficient is set to $\lambda = 1$. We evaluate our method on diffusion language models using Trado (Wang et al., 2025a) as the base model. Experiments are conducted on mathematical reasoning benchmarks including GSM8K (Cobbe et al., 2021), MATH500 (Hendrycks et al.), and AIME (Li et al., 2024), as well as coding benchmarks LiveCodeBench-V2 (Jain et al., 2024) and LiveBench (White et al., 2024). All methods are implemented within the nano-vllm (GeeeekExplorer, 2024) framework, following the same model architecture, tokenization, and decoding settings as Trado. We further evaluate our method with $\gamma = 0.85$ on video diffusion models using LongLive-1.3B (Yang et al., 2025) on the VBench2 benchmark (Zheng et al., 2025). For video generation, we adopt SpargeAttention (Zhang et al., 2025a) as the sparse attention baseline and adapt it to the block diffusion setting. Our block-external attention caching is implemented directly inside the FlashAttention (Dao et al., 2022) kernel to minimize overhead. All experiments are conducted on 4 NVIDIA A100 GPUs with a maximum batch size of 128

and a token budget of 32k.

**Main results on diffusion large language model.** We evaluate our method on Trado across mathematical reasoning and coding benchmarks. Table 1 reports accuracy on GSM8K, MATH500, and AIME, as well as pass@1 on MBPP, HumanEval, LiveCodeBench-V2, and LiveBench, under identical decoding settings, together with inference throughput measured at batch size 128. Across mathematical reasoning benchmarks, our method achieves performance closely matching dense Trado. For block size 4, accuracy differences on GSM8K, MATH500, and AIME are within 0.2%, while for block size 8, the corresponding differences remain within 1.6%. These results indicate that caching block-external attention preserves reasoning accuracy. On coding benchmarks, our method maintains comparable pass@1 to the dense baseline across different block sizes, with only minor variations depending on the task, and overall performance remains stable across MBPP, HumanEval, LiveCodeBench-V2, and LiveBench. In terms of efficiency, our method substantially improves inference throughput. At batch size 128, block-external attention caching increases throughput from 312 to 451 tokens/s for block size 4, and from 532 to 674 tokens/s for block size 8, corresponding to up to a $1.44\times$ speedup. These gains stem from reducing redundant attention computation and KV cache access across diffusion steps, and become more pronounced in long-context settings. Overall, the results demonstrate that block-external attention caching provides an effective way to accelerate diffusion language models with comparable performance.

**Latency analysis under different context lengths.** We further analyze inference efficiency by measuring per-step latency and the number of tokens participating in attention as the context length increases. Experiments are conducted on Trado with batch size 128 using two NVIDIA A100 GPUs. We vary the updated-token threshold $\tau \in \{2, 3, 4\}$, which controls when cached block-external attention can be reused without accessing the KV cache. Figure 3 presents the results. Across all values of $\tau$, our method consistently achieves lower per-step latency than the Trado baseline. As context length grows, the latency gap widens, indicating that KV cache access and attention computation increas-

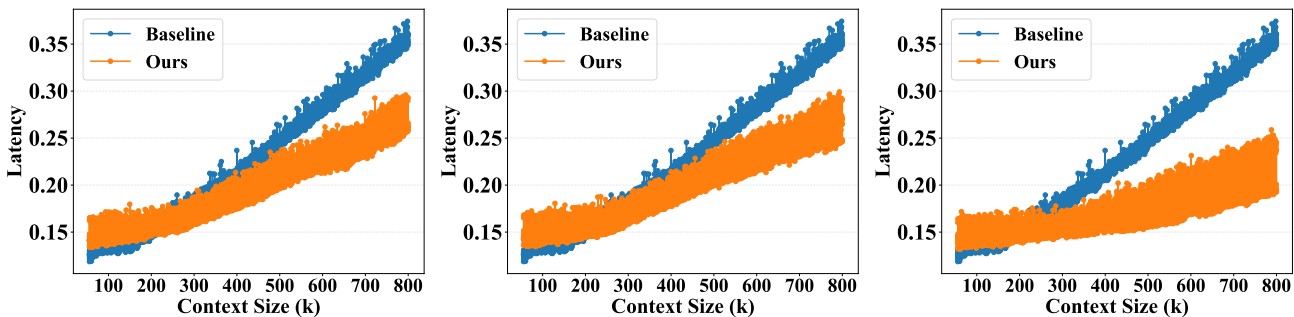

*Figure 3.* **Per-step inference latency under increasing context length.** We report results on Trado with batch size 128 using two A100 GPUs. Each column corresponds to a different updated-token threshold $\tau \in \{2, 3, 4\}$. Our method (orange) consistently reduces per-step inference latency compared to the Trado baseline (blue), with the gap widening as context length increases. Larger $\tau$ values enable more aggressive reuse of cached block-external attention, further reducing computation and memory access.

*Table 2.* Combination with sparse attention under different attention densities on diffusion language models. We report accuracy on GSM8K and MATH500, and pass@1 on HumanEval. "SparseD + Ours" applies block-external attention caching on top of SparseD.

| Method | GSM8K | | MATH500 | | HumanEval | |
|---|---|---|---|---|---|---|
| | Acc. | Δ | Acc. | Δ | Pass@1 | Δ |
| Trado-4B-Instruct | 91.21 | – | 70.80 | – | 45.12 | – |
| SparseD ($d = 20\%$) | 34.72 | +7.96 | 39.40 | +7.40 | 23.78 | +9.76 |
| SparseD + Ours ($d = 20\%$) | 42.68 | | 46.80 | | 33.54 | |
| SparseD ($d = 30\%$) | 68.61 | +3.64 | 59.20 | +5.40 | 29.88 | +6.71 |
| SparseD + Ours ($d = 30\%$) | 72.25 | | 64.60 | | 36.59 | |
| SparseD ($d = 40\%$) | 84.61 | +2.65 | 66.20 | +3.20 | 39.02 | +5.49 |
| SparseD + Ours ($d = 40\%$) | 87.26 | | 69.40 | | 44.51 | |

*Table 3.* L1 distance between sparse attention outputs and full attention for the first attention layer on a randomly sampled input. Results are reported under different attention density ratios.

| Method / Density | 50% | 40% | 30% | 20% | 10% |
|---|---|---|---|---|---|
| SparseD | 0.0014 | 0.0015 | 0.0022 | 0.0028 | 0.0031 |
| SparseD + Ours | **0.0005** | **0.0005** | **0.0006** | **0.0008** | **0.0008** |

ingly dominate inference cost in long-context regimes and that avoiding redundant access yields substantial benefits. *In particular, when the context length increases from 100k to 800k, under $\tau = 2$ the latency growth of our method is approximately half that of the baseline, indicating a theoretical speedup upper bound of up to $2\times$ in extreme long-context settings.* Larger values of $\tau$ enable more aggressive reuse of cached block-external attention and further reduce latency. The bottom row of Figure 3 shows the number of tokens participating in attention at each step. While the Trado baseline attends to the full available context, our method substantially reduces the effective attention size by reusing cached block-external attention. This reduction closely tracks the observed latency improvements, confirming that the efficiency gains primarily arise from reduced attention computation and KV cache access. Overall, these results demonstrate that our method scales favorably with context length and effectively mitigates the long-context inefficiency of block diffusion models.

**Combining with sparse attention.** Table 2 reports the results of combining our block-external attention caching with SparseD under different sparsity ratios on diffusion language models (implementation details are provided in appendix). Across all sparsity settings, integrating our method with SparseD consistently improves performance

on GSM8K, MATH500, and HumanEval. The gains are most pronounced at higher sparsity levels. For example, at a density ratio of $d = 20\%$, our method improves SparseD by +7.96 accuracy on GSM8K, +7.40 accuracy on MATH500, and +9.76 pass@1 on HumanEval. As the sparsity ratio increases, the absolute improvements gradually decrease but remain consistent, with gains of +3.64 / +5.40 / +6.71 at $d = 30\%$ and +2.65 / +3.20 / +5.49 at $d = 40\%$ on GSM8K, MATH500, and HumanEval, respectively. To better understand this effect, we further examine the deviation of sparse attention from full attention at the attention-output level. Table 3 reports the L1 distance between sparse attention outputs and full attention for the first attention layer on a randomly sampled input. As the attention density decreases, SparseD exhibits a rapidly increasing deviation from full attention. In contrast, combining SparseD with our block-external attention caching substantially reduces this gap across all sparsity levels. This analysis suggests that our method effectively compensates for the information loss introduced by aggressive sparsification by reusing stable residual attention contributions. We note that SparseD is currently implemented and evaluated only at the PyTorch level, without a sparse attention kernel compatible with paged KV cache, and therefore cannot be directly compared in end-to-end latency under vLLM-style inference.

**Aggregation and memory overhead.** The additional aggregation and memory costs introduced by `FlashBlock` are small and do not scale with context length. Table 5 shows that sparse-attention aggregation costs only about 0.07 ms across contexts, so its relative overhead decreases

*Table 4.* Video generation results on VBench2 (macro categories). We report average scores over the five major evaluation dimensions: HF (Human Fidelity), CR (Creativity), CT (Controllability), PH (Physics), and CS (Commonsense), together with end-to-end latency and attention time. Lower is better for latency metrics, and higher is better for VBench scores.

| Method | Density | E2E Lat. (s) $\downarrow$ | Attn. (s) $\downarrow$ | HF | CR | CT | PH | CS |
|---|---|---|---|---|---|---|---|---|
| LongLive-1.3B | 100% | 73.28 | 19.89 | **0.8188** | **0.1821** | 0.2796 | **0.4273** | 0.4195 |
| SpargeAttn | 30% | 72.33 | 18.12 | 0.5993 | 0.1821 | 0.2717 | 0.4019 | **0.4770** |
| SpargeAttn | 50% | 72.78 | 18.73 | 0.7609 | 0.1808 | 0.2837 | 0.4257 | 0.4080 |
| Ours | 45% | **66.07** | **12.46** | 0.7861 | 0.1818 | **0.3026** | 0.4231 | 0.4310 |

*Table 5.* Overhead analysis of `FlashBlock`. Aggregation overhead is measured when combining `FlashBlock` with sparse attention at batch size 32. Memory overhead is measured relative to the total inference footprint.

| Type | Batch | Context | Attn. (ms) | Agg. (ms) | Extra Mem. | Overhead |
|---|---|---|---|---|---|---|
| Agg. | 32 | 4k | 0.8484 | 0.0693 | – | 8.17% |
| Agg. | 32 | 8k | 1.6676 | 0.0710 | – | 4.26% |
| Agg. | 32 | 16k | 3.0385 | 0.0702 | – | 2.31% |
| Agg. | 32 | 32k | 5.9685 | 0.0717 | – | 1.20% |
| Mem. | 1 | 4k | – | – | 1.14 MiB | 0.0071% |
| Mem. | 1 | 32k | – | – | 1.14 MiB | 0.0056% |
| Mem. | 32 | 128k | – | – | 36.56 MiB | 0.1072% |
| Mem. | 32 | 1024k | – | – | 36.56 MiB | 0.0224% |

*Table 6.* Ablation on the update threshold $\tau$ and reuse-aware distillation using Trado-8B-Thinking under a 32k token budget. Dense Baseline corresponds to the original model without block-external attention caching.

| $\tau$ | Distill. | AIME | MATH500 | HumanEval | MBPP |
|---|---|---|---|---|---|
| – | – | 33.33 | 86.00 | 50.61 | 25.60 |
| 2 | $\times$ | 26.67 | 80.40 | 48.17 | 31.40 |
| 2 | $\checkmark$ | **33.33** | **85.80** | **51.22** | **33.60** |
| 3 | $\times$ | 20.00 | 78.40 | 44.51 | 30.20 |
| 3 | $\checkmark$ | **30.00** | **85.00** | **50.61** | **31.60** |
| 4 | $\times$ | 6.67 | 48.40 | 37.20 | 24.20 |
| 4 | $\checkmark$ | **13.33** | **66.20** | **42.07** | **28.60** |

from 8.17% at 4k to 1.20% at 32k. The cached block-level buffers similarly add negligible memory overhead ($\leq 0.11\%$), and the relative cost decreases as the KV cache grows.

**Effect of $\tau$ and distillation.** We conduct a joint ablation study on the update threshold $\tau$ and sparsity forcing distillation using Trado-8B-Thinking under a fixed token budget of 32k. Table 6 reports results on HumanEval, AIME, and MATH500, including the dense baseline without block-external attention caching. For each value of $\tau$, we compare models with and without distillation. Across all $\tau$ settings, reuse-aware distillation consistently improves generation quality, with particularly pronounced gains on reasoning-intensive benchmarks such as AIME and MATH500. Without distillation, performance degrades rapidly as $\tau$ increases, whereas distillation substantially mitigates this degradation. For both distilled and non-distilled models, performance remains comparable between $\tau = 2$ and $\tau = 3$, while a clear drop is observed at $\tau = 4$, indicating that overly aggressive reuse of cached block-external attention can harm quality. Figure 3 further shows that inference latency under $\tau = 2$ and $\tau = 3$ is very similar, providing little additional efficiency benefit from increasing $\tau$. Based on this quality–efficiency trade-off, we use $\tau = 2$ as the default setting.

**Performance on video generation.** Table 4 reports quantitative results on VBench2 using the LongLive-1.3B model, evaluated on five macro-level dimensions. Our method substantially reduces both end-to-end latency and attention time compared to the dense baseline, achieving an attention-time

speedup of approximately $1.6\times$ ($19.89\,\text{s} \to 12.46\,\text{s}$), while maintaining comparable generation quality across all dimensions. Compared to SpargeAttn at similar attention densities, our approach achieves higher scores on most macro categories, indicating a more favorable efficiency–quality trade-off under long-context video generation. Specially, SpargeAttn requires explicitly evaluating attention masks at each diffusion step, which introduces non-negligible overhead and limits its practical acceleration. The achievable end-to-end speedup on video generation is additionally bounded by the LongLive architecture. Specifically, LongLive employs a fixed temporal attention window of 12 frames, which limits the effective KV cache size and reduces the proportion of total inference time spent on attention computation. As a result, although block-external attention caching significantly accelerates attention, the overall end-to-end speedup remains constrained by other components of the video diffusion pipeline. Figure 4 presents qualitative comparisons on video generation, where each example is shown by a sequence of uniformly sampled frames for clear visual inspection. Videos generated by our method preserve visual quality, temporal coherence, and semantic consistency compared to the dense baseline. In contrast, SpargeAttn alone may introduce temporal inconsistencies or visual degradation in some cases. More qualitative visualizations and experimental comparisons are provided in the appendix, including additional video examples, category-wise VBench results.

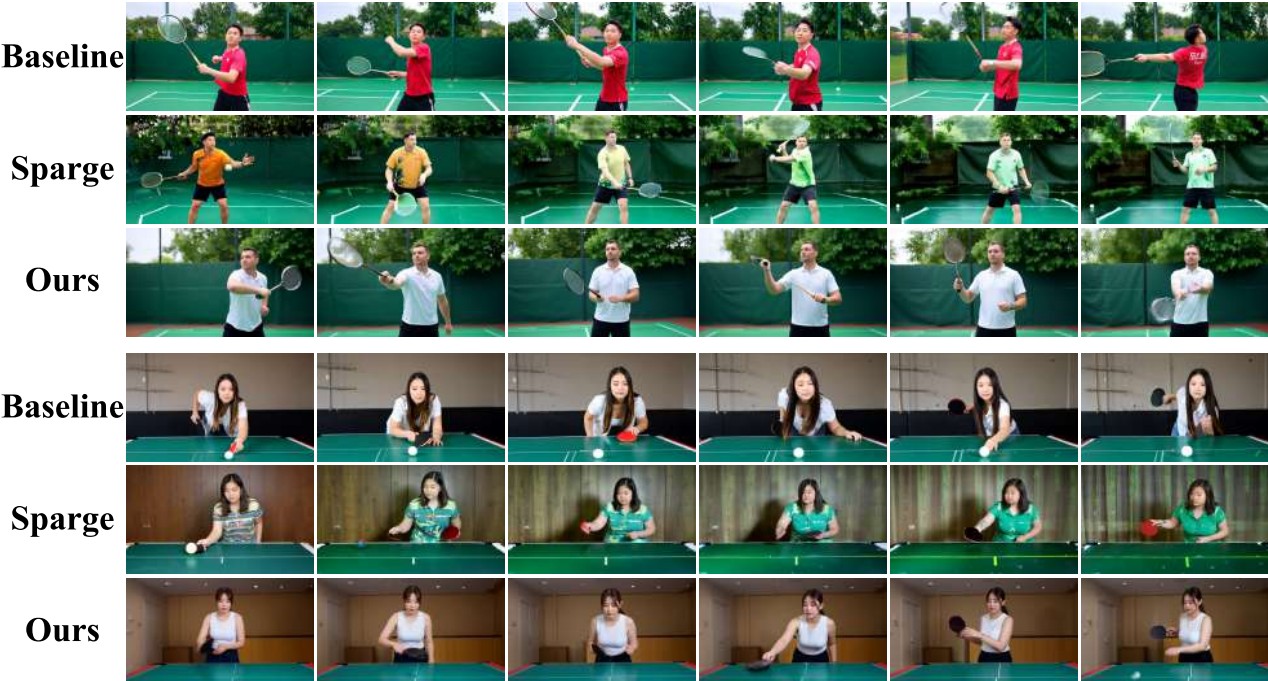

*Figure 4.* **Qualitative comparison on video generation with LongLive-1.3B.** We visualize video examples from VBench, each shown by six uniformly sampled frames. For each example, the top row shows results from the baseline, the mid row shows results from the SpargeAttention, and the bottom row shows results from SpargeAttention combined with our block-external attention caching at a fixed sparsity ratio. Our method preserves visual quality and temporal consistency while improving inference efficiency, demonstrating that block-external attention caching does not introduce perceptible degradation in generated videos.

## 6. Conclusion

We have investigated the efficiency bottleneck of block diffusion models in long-context settings and have identified cross-step redundancy in attention computation across diffusion steps. Based on this observation, we have proposed `FlashBlock`, a block-external attention caching mechanism that reuses stable attention contributions from previous steps and recomputes attention only within the current block. Experiments on diffusion language models and video diffusion models have shown that `FlashBlock` significantly improves inference efficiency. We further demonstrate that `FlashBlock` is compatible with sparse attention and can be combined as a complementary residual reuse strategy. Overall, this work highlights attention reuse across diffusion steps as an effective and complementary direction for improving long-context diffusion inference.

**Limitations and future work.** Although our method can be combined with sparse attention, we evaluate such combinations only in terms of accuracy under different sparsity ratios. Most existing sparse attention methods are not implemented with kernels compatible with block diffusion and paged KV cache, which makes efficient operator-level integration nontrivial. We leave the development of block-diffusion-aware sparse attention kernels to future work.

## Impact Statement

This paper presents work whose goal is to advance the field of machine learning by improving fundamental modeling and optimization techniques. The proposed approach has the potential to positively impact future research and applications by enabling more effective and efficient learning algorithms. Such improvements may benefit a wide range of downstream tasks where machine learning methods are applied. At the same time, as with most advances in machine learning, there may be potential negative impacts if the methods are misused or deployed without appropriate consideration of their limitations, particularly in high-stakes or sensitive application domains. We do not foresee immediate harmful consequences arising directly from this work when it is used responsibly for research purposes. Overall, we encourage practitioners to consider ethical, legal, and societal implications when applying the proposed methods in real-world settings.

## Acknowledgement

This research is partially supported by the Australian Research Council Discovery Project (DP260100218) and the CCF-Baidu Open Fund.

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

# A. Additional Clarifications and Results

**Distributional mismatch and post-training overhead.** `FlashBlock` primarily targets inference efficiency by reducing redundant attention computation and KV-cache access at decoding time. For diffusion language models, progressive unmasking can introduce a distributional mismatch when block-external attention is reused across diffusion steps. As discussed in Section 4.4, we mitigate this mismatch with an optional reuse-aware distillation stage. This stage is lightweight and parameter-efficient: we train LoRA adapters for 5,000 iterations while keeping the base model weights frozen, which takes approximately three days on two NVIDIA A100 GPUs in our implementation. Thus, the core attention-reuse mechanism remains an inference-time acceleration method, while reuse-aware distillation serves as an optional post-training adaptation when higher-fidelity dLLM generation is desired.

**Additional evaluation on LLaDA-2.0.** To further examine whether block-external attention reuse generalizes beyond Trado, we evaluate `FlashBlock` on LLaDA-2.0 (Bie et al., 2025) using SGLang on LongBench. In this setting, we do not apply reuse-aware distillation and directly use training-free attention reuse. As shown in Table 7, `FlashBlock` preserves the average LongBench performance with only a small change compared with the dense baseline. At the same time, it provides consistent decoding efficiency improvements, achieving up to $1.33\times$ decoding-phase speedup on LongBench. These results indicate that training-free reuse already transfers to another large diffusion language model with minor performance degradation.

*Table 7.* Training-free evaluation of `FlashBlock` on LLaDA-2.0 using SGLang on LongBench.

| Method | LongBench Avg. ↑ | Decoding Speedup ↑ |
|---|---|---|
| LLaDA-2.0 | 48.98 | $1.00\times$ |
| LLaDA-2.0 + `FlashBlock` | 47.96 | $1.33\times$ |

**Additional evaluation breadth and cross-model results.** To better contextualize the empirical coverage of `FlashBlock`, we include additional evaluations across language and video settings. Table 8 reports LongBench results on Trado under the same reuse setting. The average score is well preserved, changing only from 34.54 to 34.20, showing that the quality-preservation trend holds on long-context language benchmarks while maintaining the efficiency gains reported above.

*Table 8.* Additional LongBench evaluation on Trado.

| Model | Method | LongBench Avg. ↑ |
|---|---|---|
| Trado (Wang et al., 2025a) | Dense | 34.54 |
| Trado | `FlashBlock` | 34.20 |

**Comparison with attention caching methods.** We additionally compare `FlashBlock` with Pyramid Attention Broadcast (PAB) (Zhao et al., 2025), an existing attention caching method for video diffusion models. PAB directly broadcasts attention outputs across denoising steps, whereas `FlashBlock` is tailored to block diffusion and separates block-internal and block-external attention before reuse. This distinction is important because block-internal attention changes more substantially as the current block is updated, while block-external attention is more stable across steps. We therefore reuse only the more stable block-external component to reduce error accumulation while avoiding redundant attention computation. Due to time constraints, we evaluate on five representative VBench subsets. As shown in Table 9, `FlashBlock` maintains quality close to the dense baseline and outperforms PAB in the overall quality–efficiency trade-off. Compared with the least aggressive PAB setting, `FlashBlock` reduces end-to-end latency and attention time, while avoiding the degradation observed by PAB on dimensions such as Dynamic Attribute and Material.

# B. Additional Analysis on Attention Similarity in Video Diffusion

**Block-external attention stability in video diffusion.** Figure 5 presents the attention similarity analysis for LongLive-1.3B, following the same protocol as the analysis in the main text for diffusion language models. We measure the similarity of attention outputs between adjacent diffusion steps separately for block-internal and block-external attention components, across all layers and attention heads. Under LongLive-1.3B, block-external attention exhibits consistently higher similarity across diffusion steps than block-internal attention for the majority of layers and heads. This indicates that attention over the external context is substantially more stable over time, even in the presence of complex cross-step dynamics in

*Table 9.* Comparison with PAB on five representative VBench subsets. Lower is better for latency metrics, and higher is better for VBench scores.

| Method | Dynamic Attr. | Human Identity | Motion Rat. | Composition | Material | E2E Lat. (s) ↓ | Attn. (s) ↓ |
|---|---|---|---|---|---|---|---|
| PAB (step 2) | 0.2088 | 0.7376 | 0.4310 | 0.1802 | 0.3714 | 70.93 | 17.72 |
| PAB (steps 2–3) | 0.2198 | 0.6986 | **0.4483** | **0.1832** | 0.3747 | 66.73 | 13.24 |
| PAB (steps 2–4) | 0.2857 | 0.6876 | 0.3966 | 0.1814 | 0.3906 | **64.23** | **11.02** |
| Dense | **0.3583** | **0.7663** | 0.4195 | 0.1821 | 0.5674 | 73.28 | 19.89 |
| FlashBlock | 0.3475 | 0.7593 | 0.4310 | 0.1818 | **0.6138** | 66.07 | 12.46 |

video generation. At the same time, a small subset of attention heads shows lower similarity for block-external attention, suggesting that not all heads are equally stable and motivating the head-wise selective reuse strategy described in Section 4.2. This finding—that block-external attention exhibits higher similarity across diffusion steps than block-internal attention—is consistent with diffusion language models.

## C. Fine-grained VBench Results

Table 10 presents fine-grained VBench2 results grouped by macro categories, complementing the macro-level evaluation reported in Table 4. Each subtable ((a)–(e)) corresponds to one evaluation category and reports all underlying metrics used to compute the macro scores, including results from the dense baseline, sparse attention baselines, and our method. The detailed breakdown shows that the quality trends observed in the main paper are consistent across fine-grained metrics, while providing additional insight into category-specific behaviors under different attention mechanisms.

*Table 10.* Fine-grained VBench2 results grouped by macro categories. Each subtable reports detailed evaluation metrics within one category for the LongLive-1.3B model, including dense, sparse, and our method. All scores correspond to the same evaluation protocol as Table 4.

*(a)* Commonsense

| Method | Motion Rat. | Avg. |
|---|---|---|
| Dense | 0.4195 | 0.4195 |
| Sparse (30%) | 0.4770 | 0.4770 |
| Sparse (40%) | 0.4080 | 0.4080 |
| Ours | 0.4310 | 0.4310 |

*(b)* Controllability

| Method | Dyn. Attr. | Dyn. Spatial | Motion Ord. | Interaction | Plot | Landscape | Camera | Avg. |
|---|---|---|---|---|---|---|---|---|
| Dense | 0.3583 | 0.1941 | 0.3450 | 0.2222 | 0.6767 | 0.1254 | 0.2089 | 0.2796 |
| Sparse (30%) | 0.3417 | 0.2234 | 0.3382 | 0.1717 | 0.7000 | 0.1012 | 0.1356 | 0.2717 |
| Sparse (40%) | 0.3521 | 0.2125 | 0.3382 | 0.2828 | 0.7034 | 0.1333 | 0.1678 | 0.2837 |
| Ours | 0.3475 | 0.3150 | 0.3188 | 0.2660 | 0.6867 | 0.1454 | 0.1889 | 0.3026 |

*(c)* Creativity

| Method | Diversity | Composition | Avg. |
|---|---|---|---|
| Dense | 0.5345 | 0.1821 | 0.1821 |
| Sparse (30%) | 0.5013 | 0.1821 | 0.1821 |
| Sparse (40%) | 0.5234 | 0.1808 | 0.1808 |
| Ours | 0.5132 | 0.1818 | 0.1818 |

*(d)* Physics

| Method | Therm. | Material | MV-Cons. | Avg. |
|---|---|---|---|---|
| Dense | 0.1852 | 0.5674 | 0.4273 | 0.4273 |
| Sparse (30%) | 0.2315 | 0.6043 | 0.4019 | 0.4019 |
| Sparse (40%) | 0.1482 | 0.5874 | 0.4257 | 0.4257 |
| Ours | 0.1975 | 0.6138 | 0.4231 | 0.4231 |

*(e)* Human Fidelity

| Method | Anatomy | Identity | Clothes | Avg. |
|---|---|---|---|---|
| Dense | 0.8413 | 0.7663 | 0.9406 | 0.8188 |
| Sparse (30%) | 0.8234 | 0.3133 | 0.6613 | 0.5993 |
| Sparse (40%) | 0.8421 | 0.5533 | 0.8872 | 0.7609 |
| Ours | 0.8372 | 0.7593 | 0.7861 | 0.7861 |

## D. Adapting SparseD to Block dLLM

**Limitations of SparseD under Block dLLM.** SparseD (Wang et al., 2025b) was originally proposed for standard diffusion language models, where denoising is performed on a fixed-length sequence at every diffusion step. In this setting, SparseD computes a sparse attention pattern at a designated step using full attention and reuses the same pattern across all subsequent diffusion steps. This design relies on the assumption that the set of query and key tokens remains unchanged throughout the diffusion process. This assumption does *not* hold for block diffusion models. In block diffusion, denoising proceeds block by block, and only the tokens within the current block are updated at each diffusion step, while the context grows as blocks advance. As a result, the attention context is not fixed across steps, and a sparse attention pattern computed on an earlier block cannot be directly reused when the block index changes.

**Adapting SparseD to Block Diffusion.** To adapt SparseD to block diffusion, we redefine the scope of sparse pattern reuse from the entire diffusion process to individual blocks. Specifically, for each block, we apply full attention at the first diffusion step of that block and compute the sparse attention mask following the original SparseD procedure. The resulting mask is then reused for all subsequent diffusion steps within the same block. When the block advances, a new

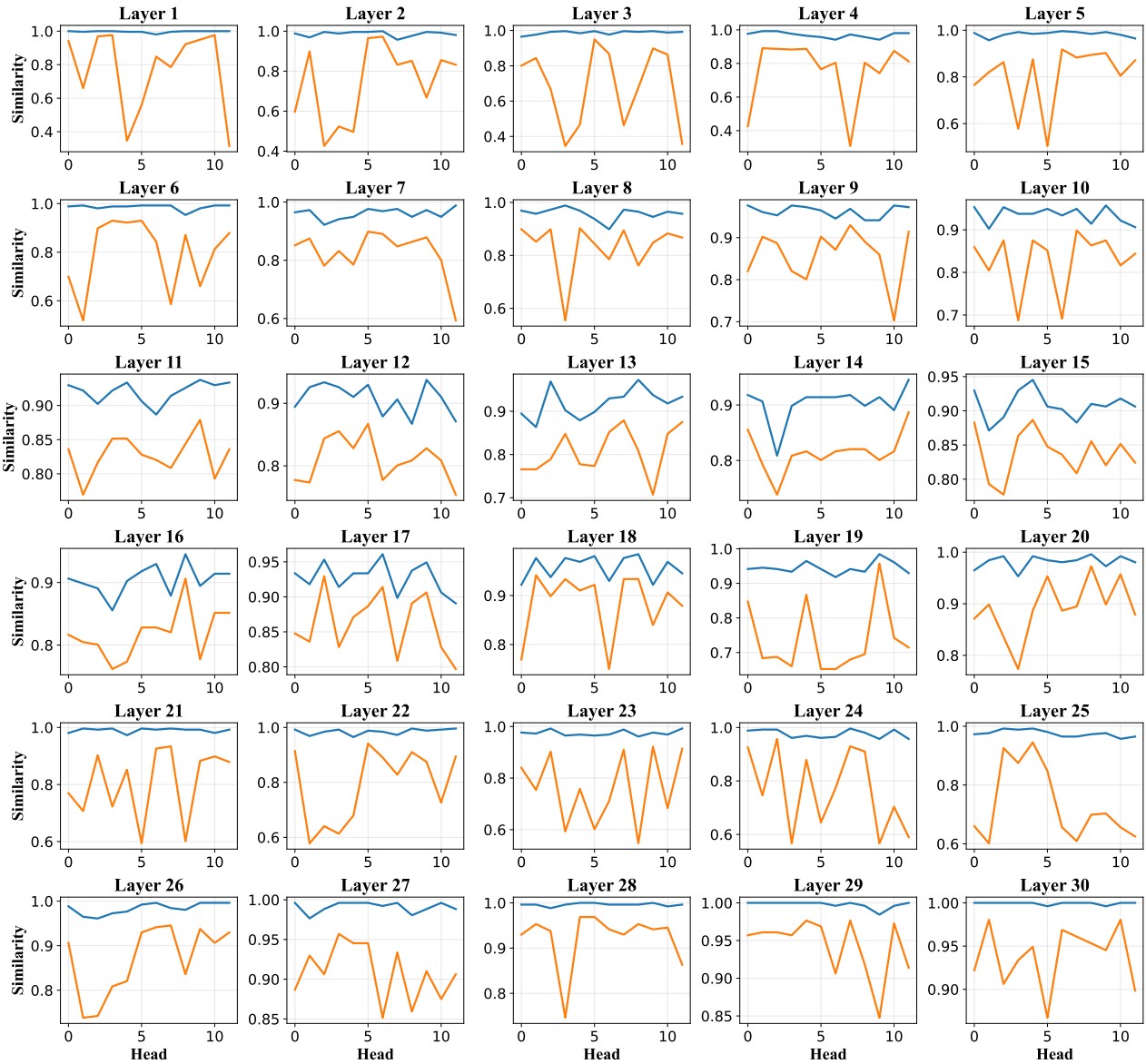

*Figure 5.* **Attention similarity across diffusion steps in video diffusion models.** We visualize the cosine similarity of attention outputs between adjacent diffusion steps for block-internal (orange) and block-external (blue) attention components across all layers and attention heads. Each subplot corresponds to one transformer layer, with the horizontal axis indexing attention heads.

sparse attention mask is recomputed for the next block using full attention.

**Combining SparseD with Attention Caching.** Figure 6 illustrates how sparse attention and block-external attention caching are combined across diffusion steps. This per-block formulation also enables a principled integration with our attention caching method. At the first diffusion step of each block, full attention is computed not only to derive the sparse attention mask, but also to obtain the attention contributions corresponding to keys that are not selected by the sparse mask. These residual attention statistics are cached at this step and reused across subsequent diffusion steps within the same block, while attention over sparse-selected keys is always recomputed. Under this design, SparseD determines which key blocks are selected by sparse attention for the current block, whereas our method determines how the attention outputs of the residual context are reused across diffusion steps. The two mechanisms operate at different levels and are therefore orthogonal, allowing them to be combined without modifying either formulation.

*Figure 6.* **Combining block-external attention caching with sparse attention.** At diffusion step $s$ (left), sparse attention selects a subset of keys (selected K/V) for explicit attention computation, while attention contributions from the remaining unselected keys are aggregated as block-external attention and cached. At the subsequent diffusion step $s+1$ (right), sparse attention is recomputed only over the selected keys, and the cached block-external attention outputs and normalizers are reused without accessing the full KV cache. The block-internal (sparse) and block-external (cached) attention components are then combined via log-space aggregation to form the full attention output. This residual reuse strategy reduces attention computation across diffusion steps and mitigates the information loss introduced by aggressive sparsification.

# E. Additional Qualitative Results on Video Generation

We provide additional qualitative comparisons on video generation using the LongLive-1.3B model to complement the quantitative results reported in the main paper. We visualize baseline generations and generations accelerated by our block-external attention caching on four representative dimensions from VBench: Motion Rationality, Material, Dynamic Attribute, and Complex Landscape. For each dimension, we show three representative video examples, where each video is illustrated by uniformly sampled frames.

Across all dimensions, our method preserves visual fidelity, temporal coherence, and semantic consistency compared to the dense baseline. Despite significantly reducing attention computation, we do not observe noticeable artifacts, motion inconsistency, or semantic drift introduced by attention reuse. These results further support our empirical finding that block-external attention exhibits strong temporal stability across diffusion steps, making it amenable to safe reuse in video diffusion.

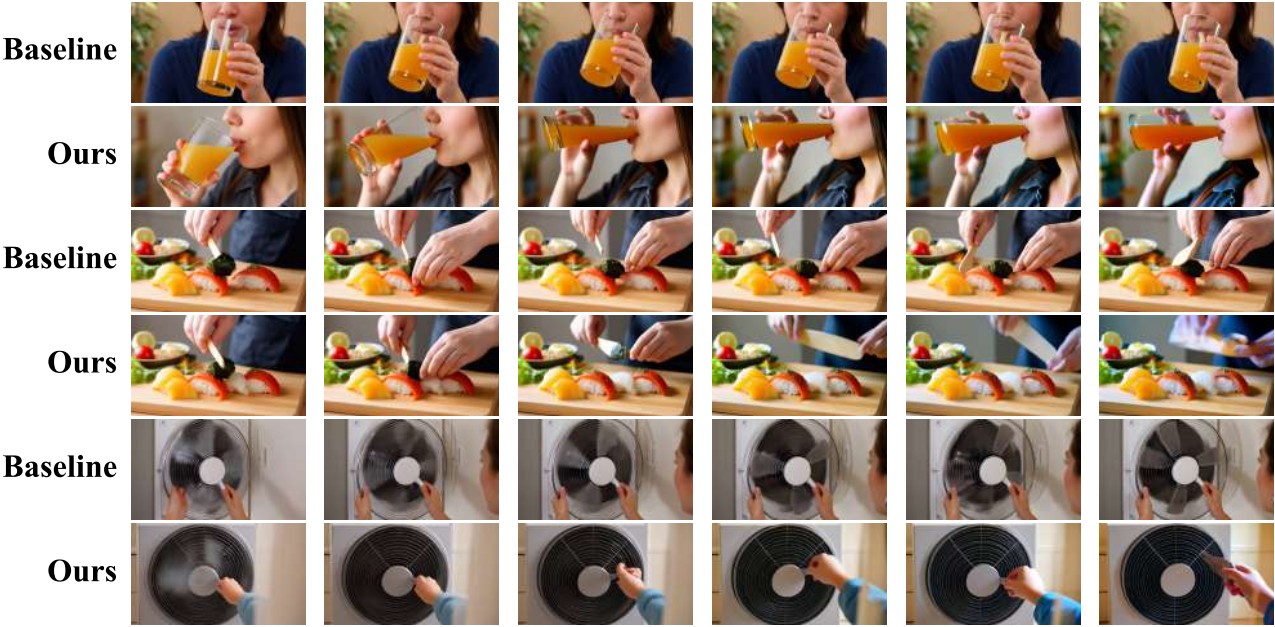

*Figure 7.* **Qualitative examples on Motion Rationality.** We visualize three representative video examples selected from the Motion Rationality dimension of VBench using the LongLive-1.3B model. For each example, we show uniformly sampled frames. Rows are organized in pairs, where the upper row corresponds to the dense baseline and the lower row corresponds to our accelerated method.

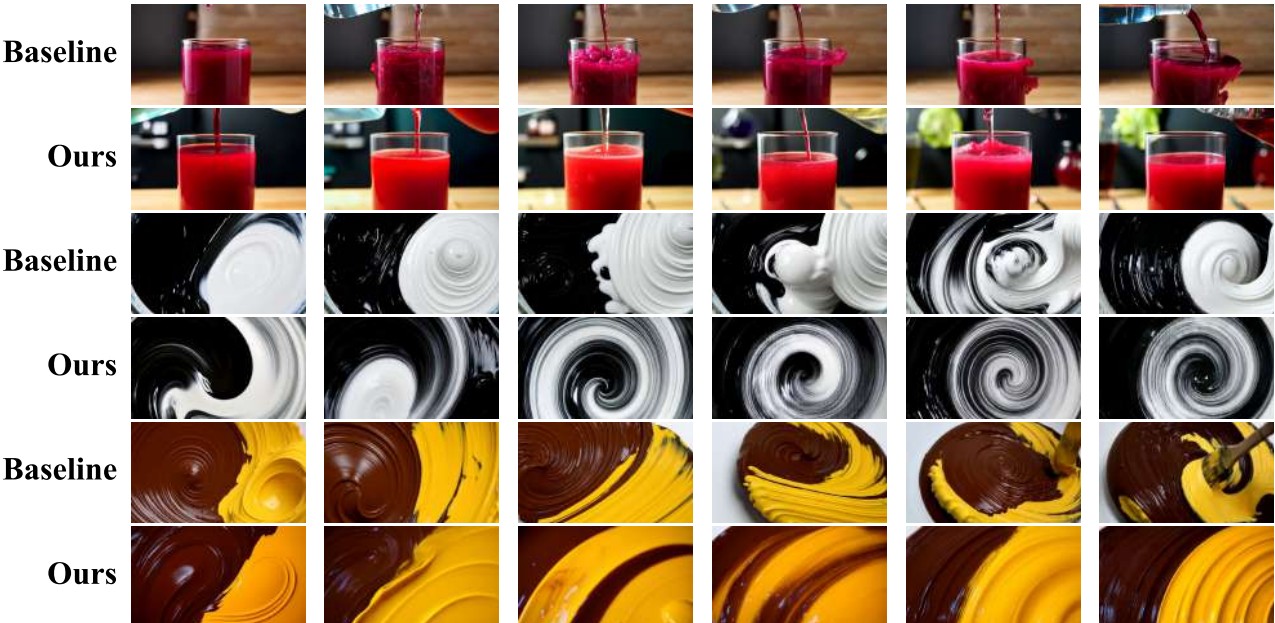

*Figure 8.* **Qualitative examples on Material.** We visualize three representative video examples selected from the Material dimension of VBench using the LongLive-1.3B model. For each example, we show uniformly sampled frames. Rows are organized in pairs, where the upper row corresponds to the dense baseline and the lower row corresponds to our accelerated method.

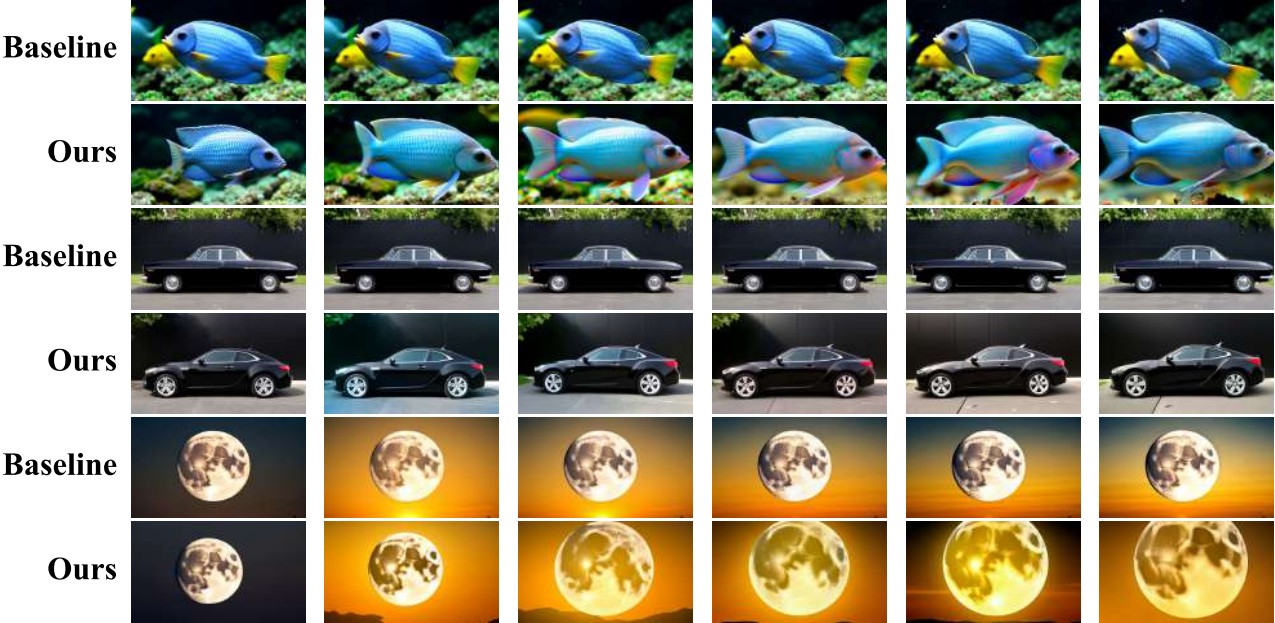

*Figure 9.* **Qualitative examples on Dynamic Attribute.** We visualize three representative video examples selected from the Dynamic Attribute dimension of VBench using the LongLive-1.3B model. For each example, we show uniformly sampled frames. Rows are organized in pairs, where the upper row corresponds to the dense baseline and the lower row corresponds to our accelerated method.

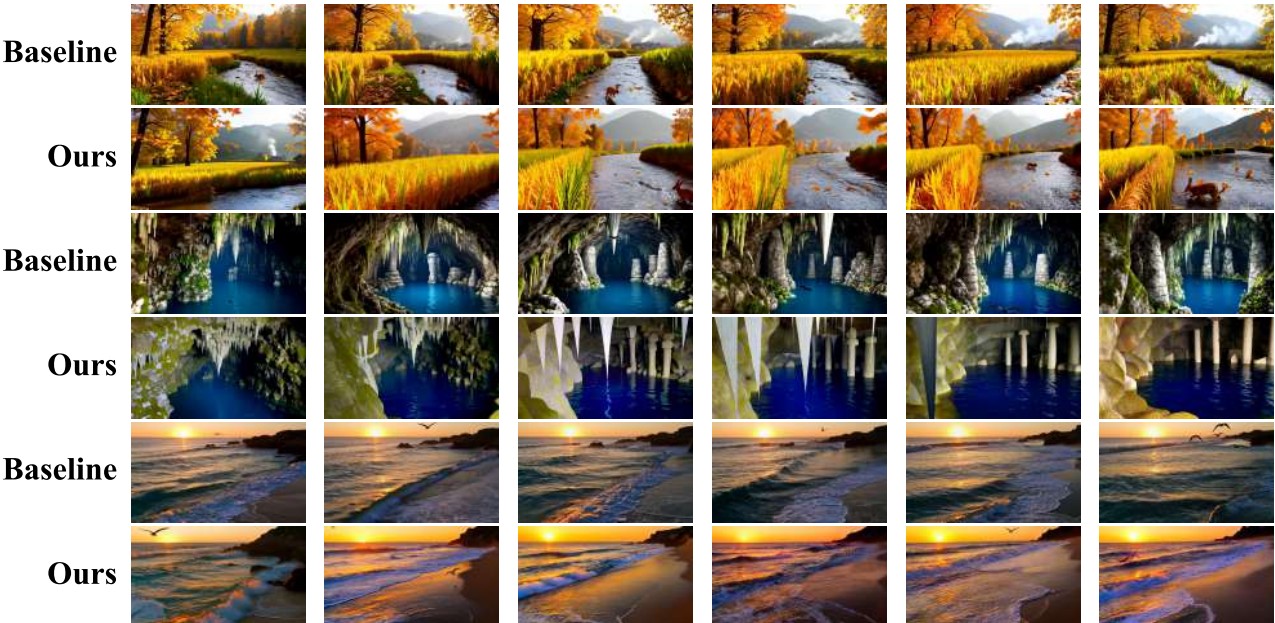

*Figure 10.* **Qualitative examples on Complex Landscape.** We visualize three representative video examples selected from the Complex Landscape dimension of VBench using the LongLive-1.3B model. For each example, we show uniformly sampled frames. Rows are organized in pairs, where the upper row corresponds to the dense baseline and the lower row corresponds to our accelerated method.

