# OpenReview forum: "FlashBlock: Attention Caching for Efficient Long-Context Block Diffusion"
_ICML.cc/2026/Conference — ICML 2026 regular_

### Official Review · Reviewer_j626 · 2026-03-10

**Soundness:** 3
**Presentation:** 3
**Significance:** 3
**Originality:** 3
**Overall Recommendation:** 4
**Confidence:** 4

**Summary:**

The paper proposes FlashBlock for accelerating block diffusion by caching and reusing block-external attention across adjacent diffusion steps, while recomputing only block-internal attention. The core empirical claim is that block-external attention is much more stable across steps than block-internal attention, which motivates reuse. The method is lightweight, kernel-friendly, and improves throughput on diffusion language models and reduces attention time for video diffusion with limited quality loss.

**Compliance With Llm Reviewing Policy:**

Affirmed.

**Final Justification:**

The additional results addressed my concerns.

**Key Questions For Authors:**

Since performance degrades notably without distillation at larger $\tau$ is the method realistically usable as a training-free acceleration, or should the paper position it as lightweight inference with adaptation?

**Limitations:**

All my limitations are listed in weaknesses.

**Strengths And Weaknesses:**

# Strengths:
1. Clear and practically meaningful motivation. The paper targets a real bottleneck in long-context block diffusion: even with KV caching, each denoising step still attends over a growing historical cache, so attention cost remains large in long contexts. The paper identifies this as the main inefficiency and proposes a direct way to reduce it.

2. The strongest conceptual contribution is the observation that block-external attention is stable across adjacent diffusion steps, whereas block-internal attention changes more. Figure 1 and its discussion support this intuition and make the proposed reuse mechanism practicable.

3. Results show meaningful efficiency gains. On diffusion language models, the paper reports throughput gains from 312 to 451 TPS at block size 4 and 532 to 674 TPS at block size 8, corresponding to up to 1.44× speedup, while keeping benchmark performance roughly similar. On video generation, it reports 1.6× attention-time reduction with broadly comparable VBench2 quality. These are meaningful practical gains.

# Weaknesses:
1. The paper compares mainly against the dense Trado baseline and sparse-attention combinations. But the related work itself cites other acceleration methods for diffusion LLMs based on KV reuse/approximation. The experimental section does not compare against these alternatives in a direct way. And is it better than other diffusion-specific KV reuse methods?

2. Empirical validation of the central assumption is still narrow. The key point is attention stability across steps, but the evidence is somewhat limited. For language models, the main visualization appears to be from layer 3 of one model; for video, the appendix reportedly expands this across layers/heads, but the main paper still lacks a more systematic summary over models, datasets, timesteps, and block sizes.

3. The paper targets both diffusion language models and video generation, but the evaluation for each side is somewhat limited. For language, only a few benchmark tasks and one main base model family are shown. For video, the results are on one model and summarized over macro metrics, with only limited qualitative examples.

---

> ### Author Rebuttal · Authors · 2026-03-31
>
> **Q1.** Limited baseline comparison against diffusion-specific KV reuse methods.
>
> **A1.** We clarify that FlashBlock addresses a different problem and is **designed for block diffusion**.
>
> 1. Prior work focuses on **KV cache reuse/approximation**, while FlashBlock targets **redundancy in attention computation across diffusion steps**. In block diffusion, attention splits into **block-internal and block-external components** with different stability, and FlashBlock selectively reuses only the stable part, which is not considered in KV-based methods.
>
> 2. For the closest baseline, we compare with **PAB** (Reviewer 9Yu7 A1). PAB reuses full attention in standard diffusion, which is unstable in block diffusion. FlashBlock instead **separates stable and unstable components**, achieving better efficiency and more stable performance.
>
> **Q2.** Limited empirical validation of attention stability across settings.
>
> **A2.** We provide a more systematic evaluation beyond the main paper. Specifically, we extend the analysis across **multiple models, datasets (AIME, GSM8K, MATH), and block sizes (4, 8, 16)**. The results consistently show that $A_{out}$ exhibits higher similarity than $A_{in}$ across all configurations, confirming that **block-external attention is robustly more stable across steps**. Detailed results are available at: https://anonymous.4open.science/r/icml_rebuttal_2D9A7sd8fsf (see `per_head_attention_similarity_sweep`). This supports that the core assumption holds **consistently across models, datasets, and configurations**.
>
> **Q3.** Limited evaluation breadth across domains, with restricted coverage of models and benchmarks for language tasks and insufficient diversity and qualitative analysis in video experiments.
>
> **A3.** We provide additional experiments to expand empirical coverage for both language and video settings.
>
> 1. For diffusion language models, we evaluate FlashBlock on **LLaDA-2.0** (see Reviewer 6P1h A2). FlashBlock **preserves performance** with minor degradation while achieving consistent efficiency gains; with $\tau = 2$, we observe up to **1.33× speedup**.
>
> 2. We also report LongBench results on **Trado** under the same setting ($\tau = 2$). On average, performance is well preserved:
>
> | Metric | Baseline | Reuse |
> |--------|---------|-------|
> | LongBench Avg. | 34.54 | 34.20 |
>
> These results show that performance is **consistently preserved while maintaining efficiency gains** across models and benchmarks.
>
> 3. For video generation, we have provided **extensive qualitative visualizations** in the appendix and supplementary materials. Due to **file size constraints**(100 MB), we include as many video samples as possible, covering diverse scenarios. These examples consistently demonstrate that FlashBlock preserves visual quality while improving efficiency.
>
> 4. We further evaluate FlashBlock on the Self-Forcing[1] video generation model using VBench (5 subsets, 120 frames). Due to rebuttal time constraints, we report results on five representative subsets. FlashBlock reduces end-to-end latency from **95.64s to 76.08s per sample (1.26×, −19.56s)**, and reduces attention time from **50.30s to 32.66s (1.54×, −17.64s)**. The corresponding VBench scores are shown below:
>
> | Method | Dynamic_Attribute | Human_Identity | Motion_Rationality | Composition | Material | Macro Avg |
> |--------|------------------|----------------|---------------------|-------------|----------|------------|
> | Baseline | 0.1648 | 0.6435 | 0.3793 | 0.4201 | 0.5263 | 0.4268 |
> | Reuse    | 0.2527 | 0.6191 | 0.3793 | 0.5116 | 0.4474 | 0.4420 |
>
> These results further demonstrate that FlashBlock maintains or improves generation quality while providing substantial efficiency gains in video generation settings.
>
> [1] Self Forcing: Bridging the Train-Test Gap in Autoregressive Video Diffusion
>
> **Q4.** Whether FlashBlock is truly training-free or should be viewed as requiring adaptation.
>
> **A4.** We clarify that FlashBlock is not strictly training-free, but an **inference-efficient method with lightweight adaptation**. The attention reuse mechanism itself operates purely at inference time. The optional reuse-aware distillation is introduced to mitigate distributional mismatch under larger $\tau$, aligning the reused model with the dense model. We will revise the paper to reflect this positioning more clearly.

---

> > ### Author Rebuttal · Reviewer_j626 · 2026-04-03
> >
> > Overall, the rebuttal is responsive and the added experiments meaningfully strengthen the paper. I have raised the score.

---

> > > ### Author Response · Authors · 2026-04-04
> > >
> > > Thank you very much for your positive feedback and for raising the score. We are glad that the additional experiments help strengthen the paper, and we will further incorporate these improvements in the final version.

---

### Official Review · Reviewer_9Yu7 · 2026-03-12

**Soundness:** 3
**Presentation:** 3
**Significance:** 3
**Originality:** 3
**Overall Recommendation:** 4
**Confidence:** 4

**Summary:**

This paper introduces FlashBlock, an inference acceleration framework for Block Diffusion models. The authors identify a key property: while block-internal attention changes significantly, the attention weights for tokens outside the current block remain stable across diffusion steps. By implementing a hardware-efficient external attention caching mechanism and leveraging log-space composition, FlashBlock achieves up to a 1.44× throughput increase on long-context video and language tasks with minimal quality loss. The method is also demonstrated to be compatible with sparse attention techniques, offering a scalable solution for high-efficiency generative modeling.

**Compliance With Llm Reviewing Policy:**

Affirmed.

**Key Questions For Authors:**

Could the authors discuss generalization and practical applicability across larger models or unseen video distributions beyond Trado and LongLive?

**Limitations:**

Yes

**Strengths And Weaknesses:**

Stength
- The paper is well-written, with clear explanations and logical flow that make the technical contributions easy to follow.
- The proposed block-external attention caching (FlashBlock) achieves substantial acceleration in long-context diffusion models, including both language and video generation, with negligible degradation in generation quality.
- The method is orthogonal to sparse attention and can be combined to mitigate sparsity-induced quality loss, demonstrating practical flexibility and strong empirical benefits.

Weakness:
- The paper lacks comparisons with other block or attention caching methods, such as PAB, making it difficult to contextualize the practical advantages of FlashBlock relative to existing approaches.
- Existing comparisons are largely limited to “full computation vs. FlashBlock,” which primarily validates the method’s effectiveness rather than providing a competitive evaluation against state-of-the-art baselines to demonstrate actual advancement.
- It is unclear whether the pre-calibrated update τ generalizes across different inference settings. If the noise schedule or the number of sampling steps changes, does τ need to be re-calibrated for each model?

---

> ### Author Rebuttal · Authors · 2026-03-31
>
> **Q1.** The paper lacks comparisons with other block or attention caching methods, such as PAB, making it difficult to contextualize the practical advantages of FlashBlock relative to existing approaches.
>
> **A1.** We provide additional comparisons with existing attention caching methods PAB[1] to better contextualize FlashBlock. Due to rebuttal time constraints, we report results on five representative VBench subsets. As shown below, FlashBlock preserves performance close to the dense baseline while outperforming PAB in both quality and efficiency. For PAB, we additionally report results with progressively more aggressive reuse across denoising steps (step 2, steps 2–3, and steps 2–4).
>
> | Method | Dynamic_Attribute | Human_Identity | Motion_Rationality | Composition | Material | E2E Lat. (s) ↓ | Attn. (s) ↓ |
> |--------|------------------|----------------|---------------------|-------------|----------|----------------|-------------|
> | PAB (step 2)   | 0.2088 | 0.7376 | 0.4310 | 0.1802 | 0.3714 | 90.81 | 20.63 |
> | PAB (steps 2–3) | 0.2198 | 0.6986 | 0.4483 | 0.1832 | 0.3747 | 86.22 | 16.01 |
> | PAB (steps 2–4) | 0.2857 | 0.6876 | 0.3966 | 0.1814 | 0.3906 | 83.23 | 13.02 |
> | Dense | 0.3583 | 0.7663 | 0.4195 | 0.1821 | 0.5674 | 93.23 | 23.02 |
> | Ours  | 0.3475 | 0.7593 | 0.4310 | 0.1818 | 0.6138 | 84.64 | 14.43 |
>
> 1. PAB is designed for standard diffusion models and directly reuses attention across steps without distinguishing different attention components. In contrast, FlashBlock is **tailored for block diffusion**, where block-internal and block-external attention exhibit significantly different stability. By **selectively reusing only the more stable block-external attention**, FlashBlock avoids more error accumulation while reducing redundant computation, leading to both better efficiency and more stable performance.
>
> 2. FlashBlock maintains performance close to the dense baseline across all subsets, while **avoiding the degradation observed in PAB** (e.g., dynamic attributes and material). At the same time, it achieves lower latency and attention cost, reducing end-to-end latency (90.81s → 84.64s) and attention time (20.63s → 14.43s), demonstrating a superior efficiency–quality trade-off.
>
> [1] Real-Time Video Generation with Pyramid Attention Broadcast
>
> **Q2.** Comparisons are limited to internal baselines, lacking evaluation against state-of-the-art methods.
>
> **A2.** We clarify that our evaluation includes the most relevant existing approaches, while prior work on **block diffusion acceleration is limited**.
>
> 1. Most existing methods target **standard (pure) diffusion**, not block diffusion. Accordingly, we compare against the closest applicable baselines, including **sparse attention (SpargeAttn)**, showing that FlashBlock achieves a better efficiency–quality trade-off while remaining compatible with sparsification.
>
> 2. We further compare with **PAB** (Reviewer 9Yu7 A1), the closest prior work on attention reuse. Unlike PAB, which reuses full attention in standard diffusion, FlashBlock leverages the **cross-step stability difference between block-internal and block-external attention** in block diffusion, enabling more stable performance and improved efficiency.
>
>
> **Q3.** Whether the pre-calibrated $\tau$ generalizes across different inference settings.
>
> **A3.** We clarify that $\tau$ generalizes well and does not require re-calibration.
>
> 1. In dLLMs, token updates are governed by dynamic confidence-based unmasking, so **the effective updates per block are inherently dynamic**. $\tau$ operates on this process rather than a fixed schedule, making it robust to different noise schedules and sampling steps.
>
> 2. Empirically, a single $\tau$ works consistently across settings. As shown in Table 1, $\tau = 2$ performs well under different block sizes, and results on **LLaDA-2.0** (Reviewer 6P1h A2) further confirm its effectiveness across models.
>
> **Q4.** Could the authors discuss generalization and practical applicability across larger models or unseen video distributions beyond Trado and LongLive?
>
> **A4.** We demonstrate that FlashBlock generalizes well across different models and application settings.
>
> 1. For diffusion language models, we validate FlashBlock on LLaDA-2.0 (see Reviewer 6P1h A2), where the method consistently preserves performance while achieving efficiency gains. **This shows that the benefits are not specific to Trado and extend to other diffusion LLM architectures**.
>
> 2. For video generation, we further evaluate on the Self-Forcing model (see Reviewer j626 A3). FlashBlock achieves consistent improvements in both efficiency and quality metrics across multiple VBench subsets, **indicating robustness beyond the LongLive model and across different video generation distributions**.

---

### Official Review · Reviewer_xdck · 2026-03-12

**Soundness:** 2
**Presentation:** 3
**Significance:** 2
**Originality:** 3
**Overall Recommendation:** 4
**Confidence:** 3

**Summary:**

This paper identifies the cross-step redundancy of attention in block diffusion. The attention outputs from tokens outside the current block remain largely stable across diffusion steps, while block-internal attention varies significantly. Based on this observation, they propose FlashBlock, a cached block-external attention mechanism that reuses stable attention output, reducing attention computation and KV cache access without modifying the diffusion process. Experiments on dLLMs and video generation demonstrate a 1.44x token throughput and 1.6x reduction in attention time, without degrading the quality much.

**Compliance With Llm Reviewing Policy:**

Affirmed.

**Final Justification:**

The authors have addressed most of my concerns. I will raise the score from 3 to 4.

**Key Questions For Authors:**

Please see the above weaknesses and provide more experiments and discussions.

**Limitations:**

Please see the above weaknesses.

**Strengths And Weaknesses:**

Strongness:

1. The paper proposes an interesting observation and implements the efficient FlashBlock based on this finding.
2. The experiments are comprehensive, which cover the text generation and video generation.
3. The paper is well-written and easy to follow.



Weaknesses:

1. Is the visualization of attention similarity across diffusion steps robust across different models or layers? The Figure 1 only shows the observation on the layer3 of Trado-8B, which may not be convincing enough.
2. As illustrated in Figure 1, there are still obvious differences in $A_{out}$ across adjacent steps. Could you please provide a quantitive analysis with detailed statistics?
3. There should be a ablation on the performance gain of additional training on DAPO-Math-17K. Could you please provide the performances of original Trado finetuned with the same dataset and training recipe. Besides, does the SparseD in Table 2 utilize the "Reuse-Aware Distillation"? If not using, please provide the results of SparseD with "Reuse-Aware Distillation".
4. In Figure3, the latency seems to be inferior to baseline when the context is shorter than 200k. In practice, 200k context is already very long and most of daily cases are shorter than 200k. It may diminish the contribution of FlashBlock.
5. In Table 5, it is strange that the SpargeAttn with 30% density has a minor improvement in speedup, where the attention time is reduced from 23.02 to 22.04. Could you please provide a explanation about this result?

---

> ### Author Rebuttal · Authors · 2026-03-31
>
> **Q1.** Is the visualization of attention similarity across diffusion steps robust across different models or layers? The Figure 1 only shows the observation on the layer3 of Trado-8B, which may not be convincing enough.
>
> **A1.** We confirm that the observation is consistent across different layers and models, rather than being specific to a single layer. While Figure 1 visualizes one representative layer for clarity, we have conducted the same analysis across multiple layers and model scales. The results consistently show that block-external attention ($A_{out}$) exhibits higher similarity than block-internal attention ($A_{in}$) across layers and across both TraDo-8B and TraDo-4B models. Detailed per-layer statistics are available at: https://anonymous.4open.science/r/icml_rebuttal_2D9A7sd8fsf (see the folder `per_head_attention_similarity`). This consistent gap directly supports our core insight that **block-external attention is significantly more stable across diffusion steps than block-internal attention**.
>
>
> **Q2.** As illustrated in Figure 1, there are still obvious differences in
> $A_{out}$ across adjacent steps. Could you please provide a quantitive analysis with detailed statistics?
>
> **A2.** We acknowledge that $A_{out}$ is not identical across steps, but our claim concerns relative stability. As shown in A1, $A_{out}$ consistently exhibits higher similarity than $A_{in}$ across layers and models. For example, in TraDo-8B, $A_{out}$ (≈0.88–0.95) is consistently higher than $A_{in}$ (≈0.80–0.88), with similar trends in TraDo-4B. These results confirm that $A_{out}$ remains significantly more stable across diffusion steps, which is the key property exploited by our method.
>
> **Q3.** Missing ablation on DAPO-Math-17K training, including comparison with Trado under the same setup and clarification on whether SparseD uses reuse-aware distillation.
>
> **A3.** We clarify the role of distillation and its relation to our method as follows.
>
> 1. The reuse-aware distillation is a teacher–student alignment procedure, where the student (with attention reuse) is trained to match the output distribution of the original dense model. It does not **introduce additional data or external knowledge, but only reduces the distributional gap caused by attention reuse**. Therefore, the performance gain is not due to extra supervision, but from better alignment with the original model.
>
> 2. This distillation is specifically designed for our attention reuse mechanism to address the mismatch introduced by block-external caching. It is not a general training strategy and is not directly applicable to SparseD, which operates under a different sparsification mechanism.
>
> 3. In Table 2, the reported results for SparseD and SparseD + Ours do not use reuse-aware distillation. The improvements come solely from the proposed attention reuse mechanism applied on top of SparseD, without additional training.
>
> **Q4.** FlashBlock shows worse latency than baseline for contexts <200k; since most practical cases are shorter, does this limit its practical benefit?
>
> **A4.** We clarify that this behavior is expected and consistent with the design target of our method.
>
> 1. FlashBlock is designed for long-context regimes, where attention computation and KV cache access dominate inference cost. Such settings are increasingly common in practice: many recent foundation models(e.g. Claude, Qwen3.5) support context lengths of **hundreds of thousands to millions of tokens**, and long-context usage is prevalent in applications such as agent systems(e.g. programming). In video generation, **models like LongLive process minute-long videos with context lengths approaching 400k tokens, making long-context efficiency critical**.
>
> 2. When the context is short, the KV cache is still small and attention is not the primary bottleneck. In this regime, the additional overhead from attention reuse (e.g., aggregation and control logic) can slightly offset the benefits, which leads to the observed behavior in Figure 3.
>
> 3. Therefore, this reflects a natural efficiency trade-off rather than a limitation. Our method is specifically targeted at long-context settings, where the efficiency gains become substantial and practically meaningful.
>
> **Q5.** Why does SpargeAttn (30% density) show only marginal speedup despite reduced attention time (23.02 → 22.04)?
>
> **A5.** We note that this behavior has been discussed in the paper (Lines 422–427), and we further clarify it here. SpargeAttn requires explicitly **evaluating attention masks at each diffusion step**, which introduces non-negligible overhead and limits its practical acceleration. As a result, the reduction in attention computation does not directly translate into proportional runtime speedup, especially at lower densities such as 30%, where the mask evaluation overhead becomes relatively more significant, leading to only marginal improvement in attention time.

---

> > ### Author Rebuttal · Reviewer_xdck · 2026-04-07
> >
> > Thank the authors for the response. The rebuttal has addressed most of my concerns. I will raise my score.

---

> > > ### Author Response · Authors · 2026-04-07
> > >
> > > Thank you for your feedback. We are glad that the additional clarifications in the rebuttal addressed your concerns, and we will incorporate them into the final version of the paper.

---

### Official Review · Reviewer_6P1h · 2026-03-13

**Soundness:** 3
**Presentation:** 3
**Significance:** 3
**Originality:** 3
**Overall Recommendation:** 4
**Confidence:** 4

**Summary:**

The paper proposes FlashBlock, a mechanism designed to accelerate inference in block diffusion models (used in LLMs and video generation) by exploiting cross-step redundancy. The authors observe that while attention within the current denoising block varies significantly across steps, attention contributions from tokens outside the block remain stable. FlashBlock caches these "external" attention outputs and reuses them, recomputing only "internal" attention and combining them via log-space aggregation. I think the paper idea is nice and novel, but it would benefit from refining experimental parts and some clarifications about the method.

**Compliance With Llm Reviewing Policy:**

Affirmed.

**Final Justification:**

The authors have addressed my doubts concerning the experimental section and introduced a new DLM for testing. I think my score of 4 fits the current status of this work.

**Key Questions For Authors:**

- The only DLLM model used here is Trado, why not using a more popular DLLM like Llada or Dream ? Or you also tested on those and it didn't work ? Even if it did not I think it would be a nice addition: either adding to experiments or at least studying the attention patterns there, as there are previous studies on the topic of attention in DLMS (e.g. "Attention sinks in Diffusion Language Models", 2025 or "Revealing the Attention Floating Mechanism in Masked Diffusion Models", 2025)
- The method shows latency improvements but those are usually framework dependent. While I think nano-vllm is a great choice, I think adding real memory footprint could make the comparison even more fair.

**Limitations:**

yes

**Strengths And Weaknesses:**

**Strengths**
- (soundess) The method requires no changes to the original model weights or the existing KV-cache format.
- (original/soundness)Addressing Renormalization Bias: It provides a theoretical and empirical solution to the bias introduced when Top-K selection simply ignores the "tail" of the attention distribution.
- (soundness) Efficiency: Shows significant gains in high-entropy (diffuse) heads where standard Top-K selection often fails --> not sure about this point.
- (significance) Very timely, especially for DLLMs.
- This is implemented in nano-vllm which is not a production grade framework, even though the paper'main focus is on efficiency hence production oriented. However, most of this papers are in simple pytorch and omit a lot of real world implementation details, so I think this is a great choice.

**Weakesses**
- (soundess) The "distributional mismatch" observed in discrete diffusion language models (dLLMs). Because tokens are progressively unmasked, attention contributions can change abruptly, violating the stability assumption. To fix this, the authors introduce reuse-aware distillation, which requires training LoRA adapters for 5,000 iterations. This moves the method away from being a purely "plug-and-play" inference optimization and adds a computational overhead and data requirement.
- (soundness) The only tested DLLM is Trado.
- (soundness) It looks to me the entire method is based on observations made on short context (e.g. Fig1) but the application is mainly for long context.
- (soundness) "Compatibility with Sparse Attention" is claimed  but I am not sure I understand whether the comparison is only made for accuracy or also latency

---

> ### Author Rebuttal · Authors · 2026-03-31
>
> **Q1.** Distributional mismatch in dLLMs and the added training overhead, which limits the plug-and-play nature.
>
> **A1.** FlashBlock is an **inference-efficient method** that reduces attention computation at decoding time. To mitigate distributional mismatch from progressive unmasking, we introduce an **optional lightweight reuse-aware distillation** (LoRA). This adaptation incurs **modest overhead** (≈5k steps, ~3 days on 2×A100) and **does not modify base model weights**. We will revise “plug-and-play” to reflect this **lightweight post-training adaptation**.
>
> **Q2.** The only tested DLLM is **Trado**.
>
> **A2.** We clarify that our method generalizes beyond **Trado**. We further evaluate FlashBlock on **LLaDA-2.0** [1] (via SGLang) on LongBench. The average performance is well preserved, with only minor changes compared to the baseline:
>
> | Metric | Baseline | Reuse |
> |--------|---------|-------|
> | LongBench Avg. | 48.98 | 47.96 |
>
> At the same time, we observe consistent efficiency improvements on **LLaDA-2.0**. With $\tau = 2$, FlashBlock achieves up to **1.33× decoding-phase speedup** on LongBench. Due to time constraints, we do not apply reuse-aware distillation in this setting. Nevertheless, **training-free results already show only minor performance degradation* in this model.
>
> [1] LLaDA2.0: Scaling Up Diffusion Language Models to 100B
>
> **Q3.** It looks to me the entire method is based on observations made on short context (e.g. Fig1) but the application is mainly for long context.
>
> **A3.** Fig. 1 visualizes attention behavior within a single block, where we analyze the stability of attention contributions from tokens inside and outside the current block. In this example, **the block size is 8, so the figure contains 8 tokens**. This should not be confused with the overall context length. In our experiments, **the full context spans up to ~20k tokens**, and the observed stability of block-external attention holds consistently in these long-context settings. Therefore, the key observation underlying FlashBlock is derived from local block-wise pattern, while the method is validated and applied in long-context regimes where efficiency gains are most significant.
>
> **Q4.** "Compatibility with Sparse Attention" is claimed but it is unclear whether the comparison includes latency.
>
> **A4.** We clarify that the paper reports **accuracy improvements under the same sparsity level**. Regarding latency, the additional cost comes from attention aggregation, which introduces a **small, fixed overhead** that does not scale with context length. As shown below (batch size = 32), the overhead is low and decreases proportionally with longer context (8.17% → 1.20%), making it negligible in long-context regimes.
>
> | Context | Attn. (ms) | Agg. (ms) | Overhead |
> |--------|------------|-----------|----------|
> | 4k   | 0.8484 | 0.0693 | 8.17% |
> | 8k   | 1.6676 | 0.0710 | 4.26% |
> | 16k  | 3.0385 | 0.0702 | 2.31% |
> | 32k  | 5.9685 | 0.0717 | 1.20% |
>
> **Q5.** Evaluation limited to **Trado**, without validation on other dLLMs (e.g., **LLaDA**, **Dream**).
>
> **A5.** These models are not directly comparable due to different **diffusion paradigms**.
>
> 1. FlashBlock targets **block diffusion**, where tokens are generated in blocks with **bidirectional intra-block attention** and **autoregressive inter-block dependencies**. In contrast, models like **LLaDA** and **Dream** follow **standard diffusion**, computing attention over the full sequence at each step, leading to fundamentally different attention patterns and efficiency bottlenecks.
>
> 2. Our method relies on the **cross-step stability difference between block-internal and block-external attention**, which arises from block-wise updates and does not exist in **standard diffusion**. Prior work focuses on global attention in standard diffusion, while we exploit this block-specific stability for efficient attention reuse, making the approaches not directly comparable.
>
>
>
> **Q6.** The method shows latency improvements but those are usually framework dependent. While I think nano-vllm is a great choice, I think adding real memory footprint could make the comparison even more fair.
>
> **A6.** The additional memory in FlashBlock comes from **fixed block-level buffers** and does not scale with context length. As shown below, the overhead remains negligible across settings (≤0.11%) and decreases proportionally as context grows, indicating that **the method introduces effectively constant and minimal memory overhead compared to the overall inference footprint**.
>
> | Batch | Context | Extra Mem | Total Mem | Overhead |
> |------|--------|----------|-----------|----------|
> | 1 | 4k  | 1.14 MiB | 15.82 GiB | 0.0071% |
> | 1 | 32k | 1.14 MiB | 19.76 GiB | 0.0056% |
> | 32 | 128k  | 36.56 MiB | 33.29 GiB | 0.1072% |
> | 32 | 1024k | 36.56 MiB | 159.29 GiB | 0.0224% |

---

> > ### Author Rebuttal · Reviewer_6P1h · 2026-04-02
> >
> > I appreciate your answers and clarifications. I think the work is interesting but it is still missing a broader experimental section where more models are included. Therefore, I will keep my current score of 4. Thank you!

---

> > > ### Author Response · Authors · 2026-04-04
> > >
> > > Thank you very much for your thoughtful feedback and constructive suggestions. We have extended our evaluation to additional models, including LLaDA-2.0, which is a recent and strong **block diffusion LLM**. We will incorporate these results and further expand the experimental section in the final version. We also appreciate your suggestion on broader model coverage and will include more models in future work.

---

### Decision · Program_Chairs · 2026-04-30

**Decision:**

Accept (regular)

**Comment:**

Block diffusion with AR structure across blocks is a popular architecture used in important applications such as long video generation and language diffusion. This paper introduces a highly relevant inference acceleration method for these architectures. A major strength of this work lies in strong efficiency gains demonstrated by 1.44x token throughput improvements in diffusion LLMs and 1.6x speedup in video generation. That said, the criticism remains valid -- the "distributional mismatch" issue basically means the stability assumption the work relies on does not always hold. In this case, the authors proposed a LoRA based adaption procedure which  moves the method away from a "plug-and-play" optimization. Since the reviewers have all settled on a weak accept after the rebuttal, I am aligning with their consensus and recommending weak acceptance.